

# Genetic signatures of ecological diversity along an urbanization gradient

Ryan P. Kelly[1], James L. O'Donnell[1], Natalie C. Lowell[2], Andrew O. Shelton[3], Jameal F. Samhouri[3], Shannon M. Hennessey[4], Blake E. Feist[3] and Gregory D. Williams[3]

[1] School of Marine and Environmental Affairs, University of Washington, Seattle, WA, United States of America
[2] School of Aquatic and Fishery Sciences, University of Washington, Seattle, WA, United States of America
[3] Northwest Fisheries Science Center, NOAA Fisheries, Seattle, WA, United States of America
[4] Department of Integrative Biology, Oregon State University, Corvallis, OR, United States of America

## ABSTRACT

Despite decades of work in environmental science and ecology, estimating human influences on ecosystems remains challenging. This is partly due to complex chains of causation among ecosystem elements, exacerbated by the difficulty of collecting biological data at sufficient spatial, temporal, and taxonomic scales. Here, we demonstrate the utility of environmental DNA (eDNA) for quantifying associations between human land use and changes in an adjacent ecosystem. We analyze metazoan eDNA sequences from water sampled in nearshore marine eelgrass communities and assess the relationship between these ecological communities and the degree of urbanization in the surrounding watershed. Counter to conventional wisdom, we find strongly increasing richness and decreasing beta diversity with greater urbanization, and similar trends in the diversity of life histories with urbanization. We also find evidence that urbanization influences nearshore communities at local (hundreds of meters) rather than regional (tens of km) scales. Given that different survey methods sample different components of an ecosystem, we then discuss the advantages of eDNA—which we use here to detect hundreds of taxa simultaneously—as a complement to traditional ecological sampling, particularly in the context of broad ecological assessments where exhaustive manual sampling is impractical. Genetic data are a powerful means of uncovering human-ecosystem interactions that might otherwise remain hidden; nevertheless, no sampling method reveals the whole of a biological community.

Corresponding author
Ryan P. Kelly, rpkelly@uw.edu

## INTRODUCTION

An enduring question of environmental science and ecology is how to measure the effects of human activities on nearby biological communities and ecosystems. While in some cases such impacts are so obvious that in-depth sampling is unnecessary to reveal them—such as paving over a wetland or clear-cutting a rainforest—many human activities are likely to have more subtle effects on the surrounding system. More adequately measuring human impacts is a core challenge as human demands on natural resources continue to grow; such measurement is a prerequisite for identifying sustainable development pathways.

The difficulty of surveying ecological communities generally results in a depth-vs.-breadth (i.e., specificity, *Rice & Rochet, 2005*) tradeoff in sampling strategy. For example, one might comprehensively survey indicator taxa with the idea that they reflect larger changes to the ecological community (*Niemi & McDonald, 2004*), or instead build limited data from many taxa into multimetric indices in an attempt to reflect some more holistic sense of ecosystem integrity (*Karr, 1981*; *Weisberg et al., 1997*). Environmental DNA (eDNA) could substantially improve upon existing survey methods by mitigating this tradeoff by providing in-depth views of ecosystems at levels of effort comparable to traditional sampling. Indeed, microbial ecology has used these same core techniques for a decade or more (*Tyson et al., 2004*; *Venter et al., 2004*; *Yutin et al., 2007*). Sequencing the diagnostic traces of genetic material in environmental samples makes it possible to detect hundreds or thousands of animals, plants, and other organisms from target habitats on ecological time scales of hours to days (*Thomsen et al., 2012*; *Turner et al., 2014*). Yet although the rapid rise of eDNA as a tool for ecological studies has featured methodological leaps and assessments of performance (*Thomsen et al., 2012*; *Ficetola et al., 2014*; *Thomsen & Willerslev, 2015*; *Evans et al., 2016*), the value that community-level eDNA methods add to traditional ecological sampling is just beginning to be apparent.

Measuring the influence of urban development on surrounding ecosystems is one application for which the broad scope of eDNA sampling may be particularly useful, in part because of the many pathways through which correlates of urbanization are likely to influence nearby ecological communities. Accordingly, it may be difficult to identify diffuse urban impacts using traditional ecological sampling alone, a particularly pressing problem as coastal urbanization increases globally (*Neumann et al., 2015*). For example, in Puget Sound, Washington, USA, as in many coastal areas, homeowners modify or harden their shorelines with concrete or other materials to protect their properties from erosion (*Scyphers, Picou & Powers, 2015*). Permitting for shoreline armoring can create conflicts between individual property rights and the communal benefits that arise from unarmored shoreline, which include storm- and flood mitigation, habitat, waterline access, and other services. Laborious manual sampling has documented some shifts in ecology as a result of shoreline armoring (*Dethier et al., 2016*; *Heerhartz et al., 2016*; *Heerhartz et al., 2014*), but the ability to detect the ecosystem effects of any stressor depends strongly upon the choice of taxa sampled. Making such informed decisions about the scope of sampling is a general problem in ecology and environmental sciences.

We assessed the effects of upland watershed urbanization on nearshore estuarine eelgrass (*Zostera marina*) communities in Puget Sound, Washington, USA using eDNA sampling at four pairs of more- and less-urban sites (Fig. 1). Puget Sound has experienced rapid urbanization over the past century, its human population increasing nearly six-fold since 1920 (*Minnesota Population Center, 2011*), and nearly four million people live within 20 km of its shore (*Bright et al., 2012*). Although preserving biogenic eelgrass habitat is now a policy priority for state and federal agencies (*Puget Sound Partnership, 2011*; *US Army Corps of Engineers, 2012*), the effect of such urbanization on eelgrass-associated fauna has been difficult to characterize with traditional sampling techniques (e.g., *Blake, Duffy & Richardson, 2014* in Chesapeake Bay). As such, the steep urbanization gradient of Puget

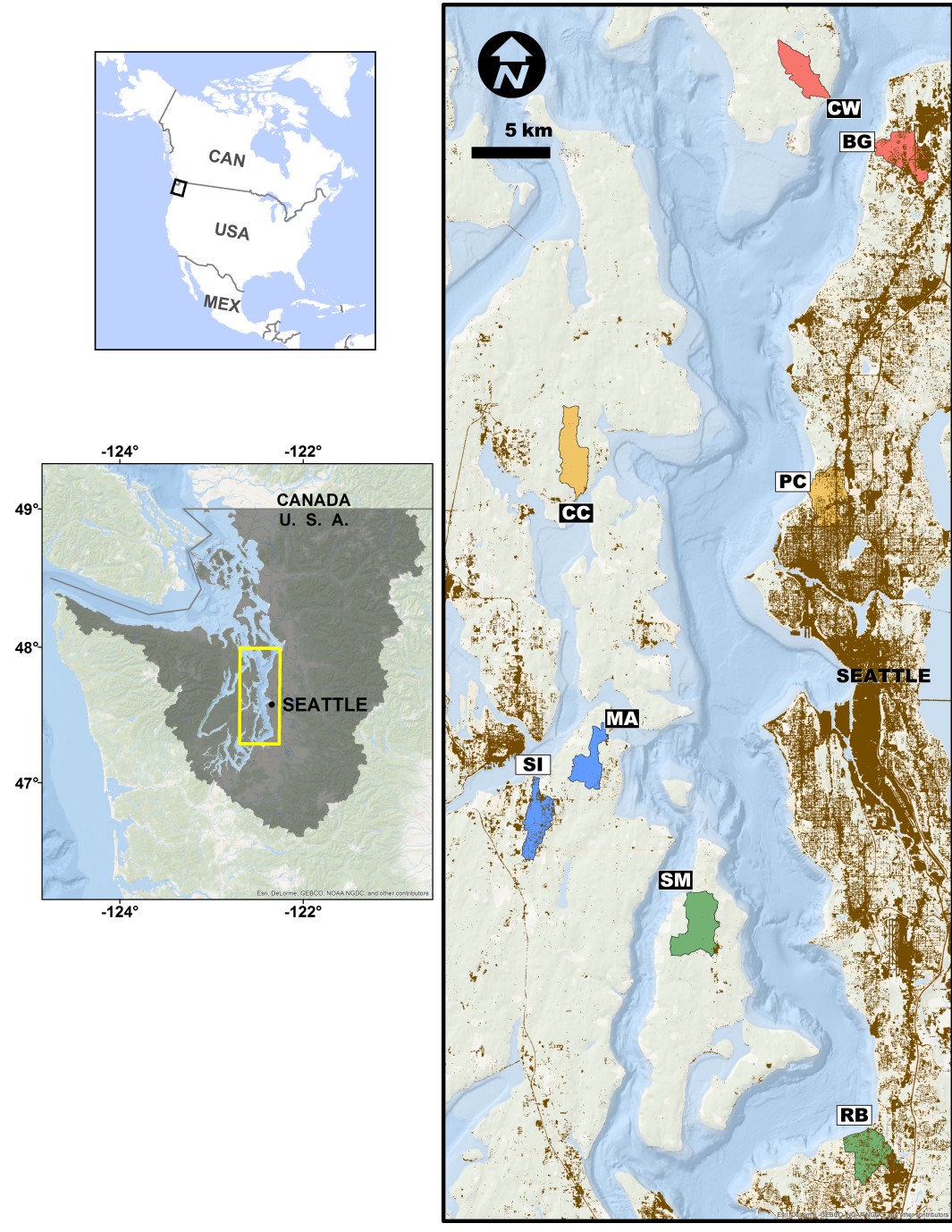

**Figure 1** **Study site sampling locations and associated stream basins.** Matched site pairs share a stream basin color. More urban sites are open boxes, less urban are black boxes. Two-letter codes correspond to site names in the 'Methods.' Brown shading indicates areas with greater than 50% area weighted-mean imperviousness.

Sound makes a compelling setting for evaluating eDNA as a means of detecting ecological differences associated with human development. Here, we report significant changes in community composition, diversity, and life-history composition associated with upland urbanization, as measured by the genetic signatures of animals detected in the water.

## METHODS

We selected eight sites in nearshore eelgrass habitats adjacent to watersheds along a gradient of urbanization in Puget Sound, Washington, USA (Fig. 1). We employed a paired study design, in which each more-urbanized site had a companion less-urbanized site at approximately the same latitude (Fig. 1), controlling for well-known geographic, oceanographic, and ecological gradients within the Sound (*Dethier, 2010*). These sampling sites included Big Gulch Creek (BG), Clearwater Casino (CC), Clinton-Whidbey (CW), Manchester (MA), Pipers Creek (PC), Redondo Beach Cold Creek (RB), Sinclair Inlet (SI), and Shingle Mill Creek (SM).

### Environmental setting

We chose sites on the basis of watershed-scale patterns of urbanization. All watershed basins were less than 1,000 ha, and contained perennial streams (*Puget Sound Nearshore Ecosystem Restoration Project, 2010*). We used three different geospatial data layers that captured various aspects of terrestrial urbanization—imperviousness (*Fry et al., 2011*), roadways (*OpenStreetMap, 2013*), and percent developed land cover (*NOAA, 2013*)—as well as percent shoreline armoring (*Puget Sound Nearshore Ecosystem Restoration Project, 2010*), to characterize urbanization at each site. Each of these individual metrics positively covaried and ordination techniques did not result in an index that was significantly more useful than any one urbanization variable alone. We therefore used imperviousness (the area-weighted mean amount of impervious surface) here as a proxy for human population and other urbanization-related parameters. This layer represents highly- to completely impermeable surfaces such as building roofs, concrete or asphalt roads and parking lots, concrete, asphalt or brick sidewalks, pedestrian walkways, and malls. We used Environmental Systems Research Institute's (Esri) ArcGIS software suite (v. 10.1) for all spatial analyses. Within site pairs, more-urban sites had higher values of imperviousness than their less-urban counterparts. Other environmental variables such as sea-surface temperature (mean, max, SD) and salinity did not systematically vary with urbanization across our sites.

### eDNA collection, extraction, and sequencing

In July 2014, we collected 1-liter water samples for eDNA analysis at each of three transects within each site, and kept these on ice until they could be processed in the lab (within hours of collection). We filtered samples onto cellulose acetate filters (47 mm diameter; 0.45 um pore size) under vacuum pressure, and preserved the filter at room temperature in Longmire's buffer following *Renshaw et al. (2015)*. Deionized water (1-liter) served as a negative control for filtering. We extracted total DNA from the filters using the phenol:chloroform:isoamyl alcohol protocol in *Renshaw et al. (2015)*, resuspended the

eluate in 200 uL water, and used 1 uL of diluted DNA extract (1:100, diluted to reduce amplification inhibition) as template for PCR. Total DNA recovered from samples (quantified using a Qubit fluorometer) was uncorrelated with site urbanization, indicating our results were not due to an accumulation of eDNA in environments near urban sites. See the Supplemental Information for additional sampling details.

We designed a novel set of primers using ecoPrimers (*Riaz et al., 2011*) to amplify approximately 114–140bp of mitochondrial 16S DNA from metazoans exclusively. These primers effectively amplify most major animal phyla—including representatives from Chordata, Arthropoda, Mollusca, Echinodermata, Nemertea, and others—while excluding non-metazoans entirely. Their sequences are as follows (5′–3′): 16s_Metazoa_fwd AGT-TACYYTAGGGATAACAGCG; 16s_Metazoa_rev CCGGTCTGAACTCAGATCAYGT.

We generated amplicons using a two-step PCR procedure, described in *O'Donnell et al. (2016)*, to avoid the taxon-specific amplification bias that results from the use of differentially indexed PCR primers (commonly used to include multiple samples onto the same high-throughput sequencing run to minimize costs). The specific PCR protocol is included in the Supplemental Information.

Each of the 24 environmental samples (3 samples/site, 8 sites) was amplified in a total of four PCR reactions, twice with each of two distinct indexed primer sets (see Supplemental Information for indexing details), for a total of $24 \times 4 = 96$ individual sets of amplicons for sequencing. All but one of the environmental samples (from site CW) was sequenced successfully. We also sequenced four positive (Tilapia; *Oreochromis niloticus* tissue) and three negative controls, treated the same way (twice with each of two indexed primers, for a total of 16 replicates of positive controls and 12 replicates of negative controls). Using tissue-derived DNA as a positive control allowed us to assess non-amplifications as deriving from sample-specific, (rather than PCR-condition-specific) causes, and selecting a non-native species as the tissue source allowed us to identify putative cross-contamination among samples (all Tilapia sequences should derive from the laboratory rather than the field). 150 bp paired-end sequencing was carried out on an Illumina Nextseq.

## Sequence processing and bioinformatics

We processed the Nextseq reads with a custom Unix-based script (*O'Donnell, 2015*), which calls existing third-party scripts to move from raw sequence data to a quality-controlled dataset of operational taxonomic units (OTUs). See the Supplemental Information for further bioinformatics details.

## Contamination removal and sequencing-depth normalization

We used a Bayesian site-occupancy modeling method to estimate the probability of the OTU representing a true positive detection (*Ficetola et al., 2014*; *Lahoz-Monfort, Guillera-Arroita & Tingley, 2016*), fitting a binomial distribution to OTU occurrences across replicates of each environmental sample, and rarefied OTUs in each sample using the smallest number of reads we observed in a single sample (124,041 reads; *Gotelli & Colwell, 2001*) to standardize estimates of taxon richness across samples. We generated 1,000 rarefied datasets, and unless otherwise specified below, we report results from one representative rarefied dataset

consisting of $11.8 \times 10^6$ reads reads representing 1,664 unique OTUs. The results do not depend significantly on the choice of rarefaction replicates; for example, replicates differed only trivially in OTU richness (mean = 1,662, sd = 9.5) and did not show different spatial trends among replicates. For beta and gamma diversity measures, in particular, OTU identity is of importance, and accordingly we show data derived from the entire set of rarefaction replicates. Finally, for each water sample, we then averaged across the four PCR replicates to estimate the abundance of each OTU. The complete eDNA dataset and analytical scripts are publicly available on Dryad (Accession: 10.5061/dryad.04tq4). See the Supplemental Information for further sequence processing details.

Our results do not depend strongly on decontamination or normalization procedures. Analyses of raw OTU data (with no decontamination or normalization), of only the most common 100 OTUs, and of only the least-common 500 OTUs, each produce the same trends in the quality-controlled and normalized data (Fig. S1). Similarly, rarefaction replicates retain the same strong trends observed in our representative single replicate (Fig. S2).

## Taxonomic annotation of eDNA sequences

We annotated the final set of OTU sequences using the command-line BLAST+ software (*Camacho et al., 2009*), searching against the complete NCBI nucleotide database (as of 12 October, 2015), with word size = 7 and up to 1,000 hits per query sequence retained. Those with no hits at $e = 10^{-13}$ (<ca. 85% identity) or better were treated as unannotated. Conflicting sequence annotations were resolved using the last common ancestor algorithm implemented in MEGAN (*Huson et al., 2011*). Disagreement among hits for a given OTU (i.e., where a single OTU is an equally good match to >1 taxon) was generally resolved at the level of taxonomic Family (83.2% of reads; Table S1).

## Data analysis, community composition, and diversity

Although amplicon sequencing produces read counts that may contain valuable information about target species abundances (*Evans et al., 2016*; *Port et al., 2016*) it remains difficult to interpret the results of amplicon studies in the context of quantitative ecology because the precise relationship between amplicon abundance and taxon abundance remains unknown and likely varies among taxa (*Evans et al., 2016*). Accordingly, our analyses used presence/absence information derived from sequence count data.

To assess the appropriateness of the spatial scale scale of sampling, we apportioned the observed variance in ecological distance (Jaccard) among sites, among transects (within sites), and among PCR replicates using a PERMANOVA. We calculated alpha diversity (=richness, or "density," sensu *Gotelli & Colwell, 2001*) at both the OTU level and at the level of taxonomic family, treating individual transects as replicates within a geographic site. We calculated beta diversity (sensu *Whittaker, 1960*, a measure of faunal change) both among transects within sites and among sites (using transect means within sites to calculate the latter), focusing on OTUs because of the loss of resolution associated with incomplete taxonomic annotation. We used Raup-Crick dissimilarity (*Chase et al., 2011*) to ensure the observed beta diversity trends were not strictly dependent upon changes in alpha diversity.

We then evaluated gamma diversity (richness across sites within a region) by generating an accumulation curve for three sets of sites: more-urban ($N = 4$ sites), less-urban ($N = 4$ sites), and all sites ($N = 8$). We sampled each set of sites (with replacement) 1000 times at each step in the accumulation curve to capture the distribution of site-specific richness.

We evaluated the relationships between diversity metrics and urbanization using linear and generalized linear regression, as well as mixed-effects models. Our data were nested, with three transect samples per site, and with each site having a single imperviousness value. To avoid pseudoreplication among transects, we used site means for linear and generalized linear regressions. For the mixed-effects models, we considered imperviousness as a fixed covariate and both site pair and site identity as a random intercept terms.

To approximate life-history diversity, we organized all OTUs for which a Family-level annotation was possible and classified each according to the following natural history attributes: Category (epifauna, infauna, demersal, pelagic, terrestrial); Habitat (terrestrial, freshwater, intertidal, subtidal); and Mobility (motile, sessile) using available reference materials such as *Kozloff (1983)*. In some cases, Families included species with a range of classifications (e.g., Cardiidae are a bivalve family which includes infaunal and epifaunal cockles found both intertidal and subtidal habitats, with a range of motility); in such cases the Family was listed as having both attributes. In all, there were 19 unique life-history niches that combinations of these attributes described (e.g., "Sessile Intertidal Epifauna," etc.; Table S2). We used these classifications to assess trends in the richness of these life-history groups with respect to imperviousness, and in a principal components analysis to assess differences in faunas among sites.

Finally, we used logistic regression and binomial tests to identify particular taxa, OTUs, and life-history characteristics significantly associated with imperviousness. We conducted all analyses in R v3.2.2 (*R Development Core Team, 2015*).

## RESULTS

Our representative rarefied eDNA (16s mtDNA) dataset recovered 1,664 operational taxonomic units (OTUs; mean of 1,000 rarefaction replicates = 1,662 OTUs ± 9.5) from a wide array of taxa characteristic of the Puget Sound estuarine environment, with 10 animal phyla represented across 27 Classes, 65 Orders, and 135 Families (Table 1). Detections included iconic groups such as *Metacarcinus* (i.e., *Cancer*) crabs, birds of prey (Accipitridae), and marine mammals (Delphinidae), with the bulk of unique OTUs reflecting molluscs (45.1%), chordates (20.2%), and arthropods (15.9%). 92% of reads (70% of OTUs) could be annotated with high confidence ($e < 10^{-32}$). These annotations included many animal taxa common to Puget Sound or the surrounding environment (Table 1; see Table S1 for full Family-level annotations).

The total variance in community-level ecological distance was attributable to differences among sites (38.6%), among transects within sites (45.4%), or among PCR replicates of the same water samples (15.9%; PERMANOVA with Jaccard distance, $p < 0.001$, 999 permutations, using OTU presence-absence data). These results are consistent with earlier work in nearshore habitats (*Port et al., 2016*), reflecting differences in eDNA profiles at spatial scales on the order of tens to hundreds of meters (here, between transects separated by ca.

**Table 1 Summary of 16s read annotations.** Summary of taxonomic annotations for 16S reads; for full annotations, see the Supplemental Information.

| Phylum | Classes | Orders | Families | Other rank |
|---|---|---|---|---|
| Mollusca | 3 | 6 | 34 | 9 |
| Arthropoda | 6 | 13 | 29 | 10 |
| Chordata | 5 | 28 | 37 | 28 |
| Bryozoa | 1 | 2 | 10 | 2 |
| Echinodermata | 5 | 8 | 13 | 4 |
| Nemertea | 3 | 2 | 8 | 0 |
| Hemichordata | 1 | 1 | 1 | 0 |
| Entoprocta | 1 | 1 | 1 | 0 |
| Porifera | 1 | 2 | 2 | 0 |

50–100 m) and limited variability due to PCR and sequencing processes. Ordination of OTU data shows transect samples largely, but not exclusively, clustering within geographic sites (Fig. S3).

## OTU diversity and urbanization

OTU richness increased significantly with upland imperviousness (Fig. 2). Family-level richness reflected the overall richness trends (Fig. 2). The results were highly robust to different decontamination or normalization procedures (Figs. S1 and S2).

Our paired sampling design controlled for potentially confounding geographically associated differences among sites. We observe the same strong positive OTU richness correlation with imperviousness in all four site pairs (Fig. 2), evidence that some aspect of urbanization—rather than confounding spatial differences among site pairs—explains the observed pattern. A mixed-effects model showed that imperviousness had a positive effect on richness after accounting for pair and site identity ($p = 0.018$).

We calculated beta diversity (faunal turnover) at two different hierarchical scales: between sites and among transects within sites. Consistent with the high level of heterogeneity, we observed among transects within sites, between-site beta diversity was uniformly high and did not differ for more- or less-urban sites (Whittaker's beta (*1960*); Wilcoxon test, $p = 0.58$). Focusing on the individual transects, however, revealed a strong decrease in within-site beta diversity with urbanization across all four site pairs: communities became more homogeneous (transects within sites became more similar) as watershed imperviousness increased (Fig. 2). Whittaker's beta (*Whittaker, 1960*) decreased from a mean of 0.816 when imperviousness was less than 10% to a mean of 0.546 when imperviousness was greater than 25% ($R^2 = 0.93$, $p$ $8.4 \times 10^{-5}$). Raup-Crick dissimilarity among transects showed a similar trend, indicating that the urbanization-associated trend in transect-to-transect variation in eDNA composition was greater than expected due to changes in alpha diversity alone.

Consistent with the trend in richness, more-urban sites had consistently higher gamma diversity than less-urban sites, as reflected in the completely non-overlapping OTU accumulation curves in those sets of sites (Fig. 2). In total, more-urban sites had 1,295 unique OTUs in 116 Families, while less-urban sites had 790 OTUs from 80 Families, respectively.

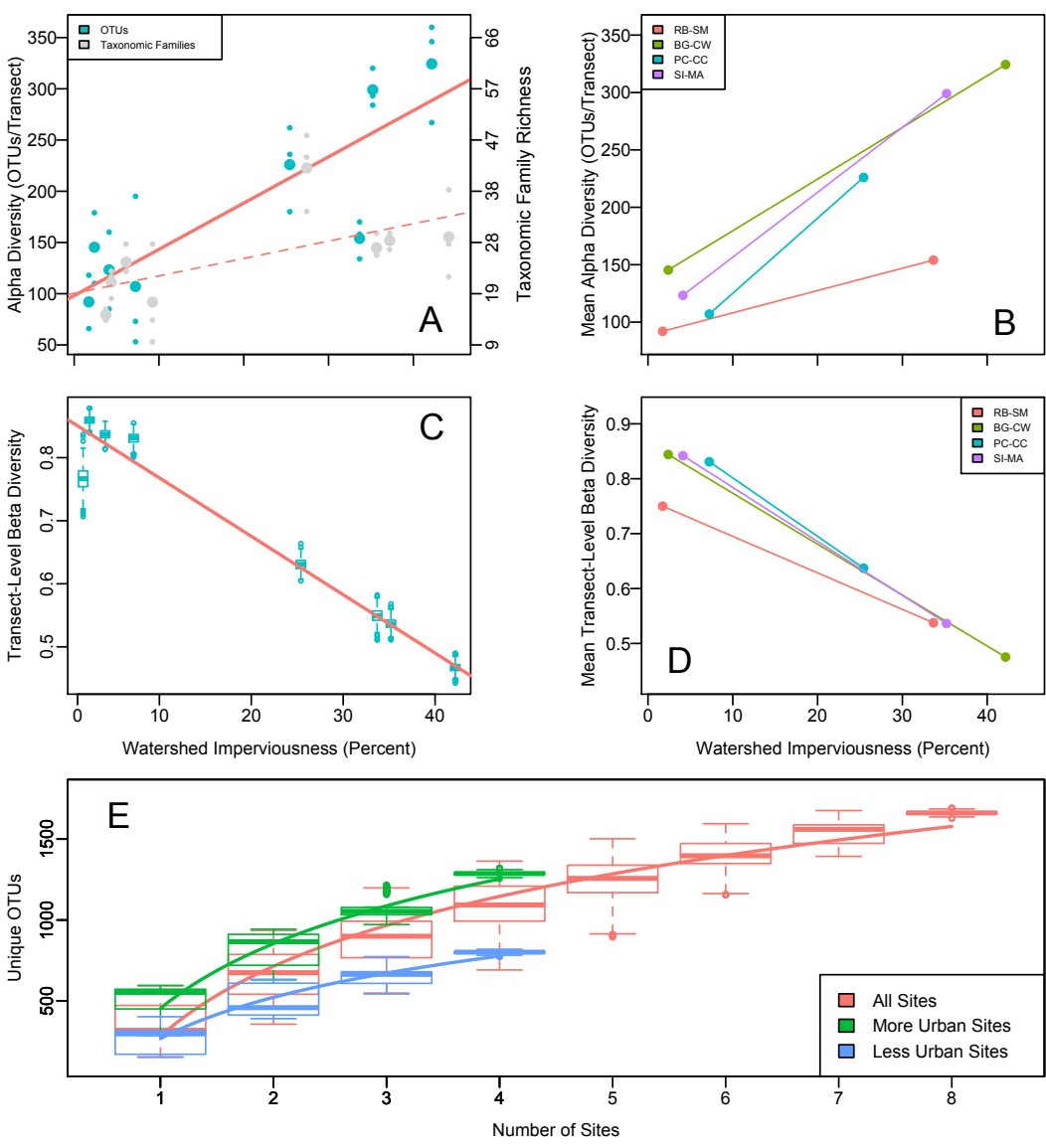

**Figure 2** **Alpha, beta, and gamma diversity recovered from water samples in Puget Sound along an urbanization gradient.** (A and B) Rarefied OTU richness and imperviousness—a proxy for urbanization—in Puget Sound. Analysis of a single focal rarefaction draw. (A) Rarefied 16s eDNA richness (solid trendline reflects OTUs; dashed trendline reflects taxonomic Families). Site means (larger circles) among transect-level data points (smaller circles). Family data shifted slightly for clarity. (B) The same data by site pair ($N = 4$ pairs of more- and less- urban sites), means plotted. Red lines indicate significant trends. Legends correspond to 2-letter site codes in Fig. 1. (C and D) (C) Mean among-transect (within-site) Whittaker's beta diversity for each of 1,000 rarefaction draws from the overall OTU dataset, rarefied to create comparable sample sizes ($N = 1.3 \times 10^5$ OTUs per transect). Linear regression on site means, $R^2 = 0.95$, $p = 3.38 \times 10^{-5}$. (D) Site means highlight the site-pair trends for single focal rarefaction draw. (E) Regional (gamma) diversity, in OTUs-per- site, as an accumulation curve. Boxplots show variance due to sampling each each set of sites (with replacement) 1,000 times from a pool of 1,000 rarefaction draws from the overall OTU dataset, rarefied to create comparable sample sizes ($N = 1.3 \times 10^5$ OTUs per transect). Best-fit logarithmic curves shown for more-urban sites ($N = 4$), less-urban sites ($N = 4$), and all sites ($N = 8$).

## Life-history diversity and urbanization

Assessing individual characteristics of habitat and mobility, taxa with differing natural history characteristics were differentially associated with urbanization. For example, OTU richness tripled with greater urbanization among sessile taxa ($p = 1.7 \times 10^{-5}$), but motile taxa increased only nonsignificantly ($p = 0.054$). Similarly, OTU richness in intertidal ($p = 7.5 \times 10^{-6}$) and subtidal ($p = 3 \times 10^{-5}$) taxa increased with imperviousness, terrestrial taxa showed no such trend ($p = 0.16$).

Community shifts among natural history types reflected richness changes by taxonomic groups. At both the OTU- and Family level, eDNA richness increased with urbanization, most notably among bivalves and gastropods (Fig. S4). Family-level bivalve richness rose, for example, from an average of five Families (37 OTUs) at <10% imperviousness to 7.4 Families (111 OTUs) at >25% imperviousness (Poisson GLM with log-link, $p < 0.01$ at family level, $10^{-16}$ at OTU level). Other taxa showed a more gradual increase in richness with imperviousness (Fig. S5), resulting in an overall increase in the number of taxonomic Families. No abundant Family declined with imperviousness.

Combining ecological characteristics into tri-variate life-history categories (e.g., "intertidal sessile epifauna") revealed 19 unique Family categories present. Life-history richness increased with urbanization (Fig. S6; $R^2 = 0.74$, $p = 0.006$), from a mean of 12.5 life histories per site in less-urban sites to a mean of 14.7 in more-urban sites, due to the concomitant increase in taxon richness at more-urban sites. Normalizing by the number of Families present at each site reveals a strong decrease in occupied life-histories-per-taxon with urbanization, from a mean of 0.66 in less-urban sites to 0.47 in more-urban sites (although the trend is nonsignficant; Fig. S6; $R^2 = 0.38$, $p = 0.1$). Ordination of the life histories results in identifiable sites and urbanization categories (Fig. S7), similar to the ordination plot for OTUs.

Beyond community measures, we identified 46 individual OTUs—again dominated by bivalves (33 OTUs from five Families)—that were positively correlated ($p < 0.01$; logistic regression) with upland imperviousness. Gastropods (five; limpets), urchins or sand dollars (seven; not classifiable to family level), and one fish OTU comprised the remaining 13 OTUs. Conversely, a single OTU was negatively correlated with imperviousness (a mytilid mussel OTU). Providing some direct indication of human influence on the nearshore Puget Sound, human OTU richness increased significantly with imperviousness ($p = 0.01$; Poisson GLM), as did richness in selected taxa cultivated commercially (*Panopea*, $p = 5 \times 10^{-4}$; *Bos*, $p = 0.005$) or introduced taxa (*Mya*, $p = 5.9 \times 10^{-6}$).

## DISCUSSION

All organisms leave behind residual genetic signatures in their environments, which provide the opportunity to explore patterns of diversity and community structure that may not be possible otherwise. Here, we recovered these signatures from nearshore estuarine habitats along an urban gradient, revealing strong trends in the diversity of animals and ecological roles present. While alpha (site richness) and gamma (regional richness) diversity strongly increased with upland urbanization, more-urban sites were significantly more homogeneous (within sites) than less-urban sites. Life-history diversity largely paralleled

these same trends, with a greater richness of ecological life histories among taxa found in more urban areas, but greater redundancy in life-history niches among these taxa. Taken together, our results suggest that more urbanized upland areas support larger suites of species, with less compositional variation, in and around downstream eelgrass habitats. Further, we find evidence that the mechanisms of land–sea interaction act at watershed scales, rather than at the larger scale of Puget Sound. These results also substantiate the idea that eDNA can be a powerful addition to traditional means of assessing human-ecosystem interactions.

## Trends in diversity and ecological function with urbanization

Although dense urban areas do not necessarily decrease biodiversity in general (*Ives et al., 2016*) and the effects of urbanization on species richness appear to be taxon- and spatial-scale-specific (*Shochat et al., 2006*), the positive richness trend we see in Puget Sound 16s eDNA is nevertheless striking. Several plausible mechanisms could explain the increase in 16s eDNA richness, although our study design prevents us from assessing causation explicitly.

One likely explanation for the trend is the interaction between fauna sampled with eDNA and the kinds of habitat that are more common near urban settlements. Our study design attempted to sample identical habitats across all sites; however, there may be unobserved differences in habitats. For example, our results may reflect an increase in availability of muddy habitats associated with urbanization, and a concomitant increase in richness within those habitat patches.

A second plausible mechanism is that greater anthropogenic nutrient inputs into urban areas yields greater productivity. Urbanization greatly increases total nitrogen fluxes into rivers and estuaries (*Rabalais et al., 2009*; *Mohamedali et al., 2011*), and increased primary productivity, which may result from such fertilization, is generally—but not strictly— associated with increased secondary productivity (*Leslie et al., 2005*) and taxonomic diversity (*Mittelbach et al., 2001*; *Whittaker & Heegaard, 2003*). However, Puget Sound, like many coastal systems, is dominated by marine derived nutrients (*Mackas & Harrison, 1997*; *Mohamedali et al., 2011*), suggesting that any fertilization effect from small watersheds such as those we focus on here is unimportant. Each of the urban sites we sampled also has a wastewater treatment facility in the vicinity. However, all outflows from treatment facilities occur in deep water offshore, far from our sampling areas, making any effect of fertilization indirect at best. Wastewater treatment facilities could also increase richness by concentrating genetic material originating elsewhere. However, although the increase in human OTUs we observe is consistent with this hypothesis, the great majority of DNA recovered stems from Puget Sound species rather than taxa likely to be dominant in human waste streams and none of our results is driven by exogenous eDNA.

Intriguingly, as eDNA communities increased in richness with urbanization, they also became more homogeneous. Others have found that increased subtidal sedimentation— associated with the kind of low-energy environments we sampled here—tended to make rocky reef communities more similar to one another (*Balata, Piazzi & Benedetti-Cecchi, 2007*), and nutrient enrichment can have the same effect in lakes (*Donohue et al., 2009*). Our

results are consistent with the idea that urbanization tends to homogenize communities even though the total number of unique taxa may increase (*Urban et al., 2006*; *Piazzi & Balata, 2008*). A similar effect is also associated with non-indigenous species introductions (*Rosenzweig, 2001*), but non-indigenous species do not drive the trends we observe here. Although a comprehensive list of native taxa is not available against which to compare our results, the annotated Families are nearly all familiar native taxa from Puget Sound; moreover, the trends we report are consistent across even small subsets of the data (Figs. S1 and S2), indicating our results do not depend upon a small set of potentially non-indigenous taxa.

More generally, beta diversity can help disentangle the ecological forces behind community assembly (*Condit et al., 2002*; *Tuomisto, Ruokolainen & Yli-Halla, 2003*; *Dornelas, Connolly & Hughes, 2006*; *Chase, 2007*; *Chase, 2010*; *Chase & Myers, 2011*), by distinguishing niche-related deterministic processes from stochastic ones. Our observations are consistent with the idea that that deterministic, possibly niche-related, processes significantly influence Puget Sound nearshore communities: transect-to-transect beta-diversity declined steadily with an environmental gradient of urbanization independent of geographic space, and per-taxon life-history richness similarly declined (albeit nonsignificantly) across this same environmental gradient.

We expect different ecological patterns to be apparent at different spatial scales, and conversely, the scales of ecological patterns provide hints about the mechanisms driving those patterns (*Levin, 1992*). Given the site- and transect-level differences we observed, it seems likely that the mechanisms mediating the human-ecosystem interactions in Puget Sound occur at the watershed scale (~100s of meters), rather than at larger scales of urbanization (e.g., Puget Sound scale, 10s of km). Urbanization does not appear to homogenize communities across sites; more-urban sites were just as different from one another as less-urban sites were, and the gamma diversity accumulation curve indicated that additional urbanized sites continued to feature new OTUs. The real differences associated with urbanization occurred within sites, with more-urban sites being more homogeneous (i.e., smaller differences among transects) than less-urban sites. In sum, we did not observe a generalized "urban" fauna at urban sites. Instead, each urbanized site had a distinct ecological community, exhibiting greater richness, lower spatial variability, and greater life-history redundancy than a similar less-urban site, but without a shared, characteristic community.

Regardless of the precise mechanism, the eDNA data reveal a strong signal of land-sea interaction (*Samhouri & Levin, 2012*). Especially in light of ever-increasing human population density in coastal areas worldwide (*Neumann et al., 2015*), our results suggest that eDNA can be a powerful tool for uncovering human-ecosystem interactions that might otherwise remain hidden.

## eDNA as an emerging tool for ecological analysis: scale and selectivity

Ecology and related disciplines depend upon techniques to sample and describe communities, ecosystems, and their properties. However, any one set of samples yields a necessarily biased view of the world; ten different sampling methods can yield ten different

results even with small numbers of target taxa (*Valentini et al., 2016*). This selectivity is usually intentional—e.g., settlement plates are designed to sample bryozoans rather than seals—but where unintentional, such selectivity can bias results in ways that often remain unexplored (*Baker et al., 2016*).

The rise of eDNA sampling has led to studies comparing molecular techniques either to traditional methods or to known communities. Single-taxon qPCR studies have compared favorably with traditional surveys in terms of detection rates (*Jerde et al., 2011*; *Takahara et al., 2012*; *Eichmiller, Bajer & Sorensen, 2014*; *Laramie, Pilliod & Goldberg, 2015*), with sequence-based (i.e., metabarcoding) analyses proving more difficult to interpret relative to traditional sampling, in part because of difficulty of comparing detection rates across methods (*Cowart et al., 2015*). eDNA is an in-depth sampling technique that yields interesting and repeatable results; however, the absence of eDNA detection does not imply absence of taxon of interest (*Roussel et al., 2015*). One eDNA locus, or even several loci, will not reveal all of the taxa present in an area. Indeed, eDNA sampling with a different genetic locus—or even a different set of primers at the same locus—would have yielded a different suite of taxa (e.g., *Cowart et al., 2015*).

Consistent with earlier observations from a study of *Zostera* communities (*Cowart et al., 2015*), our single eDNA locus failed to detect epifauna known from the sampled sites. Hippolytid and crangonid shrimp, littorinid snails, idoteid isopods, and others were common in the field (JF Samhouri et al., 2016, unpublished data) but absent from the eDNA, likely due to amplification bias and primer mismatches. Such performance does not make eDNA inappropriate for biodiversity monitoring, but rather put sequenced-based sampling in the company of every other sampling technique (*Shelton et al., 2016*). Because the "true" community remains unknown (*Shelton et al., 2016*), it is impossible to evaluate error rate in an absolute sense for any field-based method. Given that nearly all of the taxa we detect here are known from local waters or the surrounding area, our false positive rate for eDNA appears to be very low. We suggest that community-level eDNA surveys be viewed in a light appropriate to any new sampling technique: biased relative to some unknown true value, but significantly complementing existing imperfect sampling techniques such as tow nets and other manual collections.

Finally, our results suggest that eDNA recovers fine-scale differences in ecological communities, such that transects tens of meters apart can be as different as transects kilometres apart. Nearly half (45%) of the variance in ecological distance was due to differences between transects at the same sampling site, consistent with the fine-grained spatial resolution reported by *Port et al. (2016)* in another nearshore eDNA amplicon study. This observation supports a growing sense that eDNA may travel only limited distances away from its sources, depending upon the environmental context (*Eichmiller, Bajer & Sorensen, 2014*; *Deiner & Altermatt, 2014*; *Laramie, Pilliod & Goldberg, 2015*), and provide further evidence that eDNA variation at small spatial scales is more likely signal than noise. Nevertheless, it is not obvious why eDNA might exhibit such variability on the order of tens of meters (in this study and in others; *Eichmiller, Bajer & Sorensen, 2014*; *Port et al., 2016*), but simultaneously feature the genetic signatures of species that are not in the immediate vicinity. Examples here include terrestrial and aquatic taxa, whose DNA must have travelled

at least some distance into the intertidal habitats sampled. One explanation is that—if genetic material is detectable as a steady-state balance of generation, degradation, advection, and diffusion away from a point source—such transportation is to be expected at low levels, even when the bulk of genetic material remains close to its source. Consistent with this model, the great majority of taxa in our data are marine, with non-marine taxa only at low levels (6% of reads including human DNA; 3% not including human DNA; see Table S1).

## CONCLUSION

Sampling using eDNA sequencing offers a breadth of taxonomic coverage valuable for both basic and applied ecology. Our results demonstrate the power of this technique for assessing human-ecosystem interactions in a nearshore environment, revealing significant trends in animal diversity and life history likely linked to human alteration of upland habitats. Like all sampling methods, eDNA offers a view of the world that is both biased and incomplete, in the sense that surveys using a given gene will detect some taxa and not others. Traditional sampling has analogous drawbacks. Here, data from a single genetic locus provided a reasonably holistic view of the Puget Sound nearshore ecosystem—encompassing taxa as diverse as high-intertidal barnacles, birds of prey, and subtidal bivalves, from a wide variety of ecologically-linked habitats—that strongly suggests urbanization has generated unexpected consequences for a large number of nearshore taxa, particularly those with sessile lifestyles. Consistent with JF Samhouri et al. (2016, unpublished data), *Blake, Duffy & Richardson (2014)* and *Ives et al. (2016)*, we see these results as a counterexample to the idea that humans uniformly decrease biodiversity. Rather, the observation that more urbanized areas support larger, but more homogeneous, suites of species indicates a more nuanced effect of human alteration on nearshore communities.

## ACKNOWLEDGEMENTS

We thank J Port, L Sassoubre, and A Boehm; A Stier, and P Levin; M Dethier, E Heery, J Toft, and J Cordell; R Morris and V Armbrust; J Kralj; A Wong, E Garrison, J Levy, M Klein, and E Buckner; coastal property owners for access to field sites; and the Helen R. Whiteley Center at Friday Harbor Laboratories, and two anonymous reviewers.

### Funding

This work was supported by a grant from the David and Lucile Packard Foundation to RPK (grant 2014-39827). The funders had no role in study design, data collection and analysis, decision to publish, or preparation of the manuscript.

### Grant Disclosures

The following grant information was disclosed by the authors:
David and Lucile Packard Foundation: 2014-39827.

## Competing Interests

The authors declare there are no competing interests.

## Author Contributions

- Ryan P. Kelly, James L. O'Donnell, Andrew O. Shelton and Jameal F. Samhouri conceived and designed the experiments, performed the experiments, analyzed the data, contributed reagents/materials/analysis tools, wrote the paper, prepared figures and/or tables, reviewed drafts of the paper.
- Natalie C. Lowell and Shannon M. Hennessey performed the experiments, contributed reagents/materials/analysis tools, prepared figures and/or tables, reviewed drafts of the paper.
- Blake E. Feist performed the experiments, analyzed the data, contributed reagents/materials/analysis tools, prepared figures and/or tables, reviewed drafts of the paper.
- Gregory D. Williams conceived and designed the experiments, performed the experiments, contributed reagents/materials/analysis tools, prepared figures and/or tables, reviewed drafts of the paper.

## DNA Deposition

The following information was supplied regarding the deposition of DNA sequences:
    BioProject ID: PRJNA338801.

## Data Availability

    Kelly RP, O'Donnell JL, Lowell NC, Shelton AO, Samhouri JF, Hennessey SM, Feist BE, Williams GD. Data from: Genetic signatures of ecological diversity along an urbanization gradient. Dryad Digital Repository: http://dx.doi.org/10.5061/dryad.04tq4.

## Supplemental Information

Supplemental information for this article can be found online at http://dx.doi.org/10.7717/peerj.2444#supplemental-information.

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
