# Peer review of "Genetic signatures of ecological diversity along an urbanization gradient"

_PeerJ, doi:10.7717/peerj.2444_

## Round 0.1 · original submission · Minor Revisions

Dear Ryan,

Both reviewers suggest Accept but have some questions and minor points that they would like you to address. I agree with the evaluations of both of the reviewers, and because of these points I made a decision of minor revision. But please consider the article accepted.

All the best,

Tomas Hrbek

Reviewer 1 ·

Basic reporting

The manuscript is clearly written and meets the basic requirements of PeerJ publications.

Experimental design

The submission describes- to my knowledge- original research which presents a clear research question and contains appropriate and rigorous methodology.

Validity of the findings

I believe the findings and conclusions made by the authors are valid.

Additional comments

I have reviewed the manuscript “Genetic signatures of ecological diversity along an urbanization gradient” by Kelly et al. The manuscript describes the application of environmental DNA (eDNA) surveillance methodology and next-generation sequencing to explore biodiversity impacts at more and less developed sites within the Puget Sound. I believe the manuscript successfully advances the field of eDNA study toward addressing more interesting and fundamental ecological questions, and the exploration of potential homogenization along a gradient of human impacted systems will also be of broad appeal to the community of ecologists. I do have several minor comments that could improve the manuscript if addressed, which I list below.
• Line 104: Please provide the methodology for how total DNA was quantified.
• Line 124: Please justify choice of a non-native fish as the positive control- to what extent does this control for what one would expect to see in environmental samples within the Puget Sound?
• Lines 141-142: Please provide some additional information to support the statement “the results do not depend significantly on the choice of rarefaction replicates.” For example, was there some sort of bootstrapping or power analysis that preceded the final results presented within the manuscript?
• Line 214: Following the data presented here, 8% of reads (30% of OTUs) could not be annotated as species common to the Puget Sound or surrounding environment. I would like to see these discussed in more detail, as I think it is relevant to the overall patterns observed (see next point).
• Line 341-342: Invasive species are presented very rapidly as a possible explanation for the results presented within the manuscript, but not explored in any detail. Unlike other explanations presented within the discussion whose causation cannot be assessed explicitly (Line 313-314), I believe invasion-mediated biotic homogenization could be explored in more detail. Relevant to my previous point, are the same introduced species present at each site? This would result in increased alpha and decreased beta diversity as observed. Thus, a table identifying non-native species/families at each site could be quite informative.

·

Basic reporting

No Comments.

Experimental design

No Comments.

Validity of the findings

No Comments.

Additional comments

In this manuscript, Kelly et al. use analysis of eDNA in the marine environment to infer effects of urbanisation on marine biodiversity using a range of ecological metrics. I thought the results were interesting, the analysis well executed and interpreted, and the manuscript was a pleasure to read. I have very little to add, but the authors may wish to consider the following suggestions:

L90. The concept of “imperviousness” is used throughout the manuscript, and while I assume it’s just the proportion of concrete and man-made hard surfaces in the environment, it might be good to define this a little better for the casual reader.

L104. Why was the DNA extract from the filters diluted 1:100?

L109. The primer bias and non-detection issue was discussed in the manuscript, but given that these are new primers and may end up being useful to the eDNA community, it might be worth running an in-silico step to determine rough degree of primer coverage over perhaps a selection of local marine species that have data on GenBank. The program MFEprimer is quite good for this (https://github.com/quwubin/MFEprimer).

L312. Another possibility to consider which might explain the results are prevailing winds or tides. Obviously these factors influence where humans settle, and there also may be an effect in consistently pushing water from areas of higher biodiversity toward or away from other areas?

---

## Round 0.2 · accepted · Accept

Dear Ryan,

Thank you for revising your MS. In my opinion you satisfactorily revised the MS and addressed all pertinent comments of the referees, and thus I am recommending accept.

All the best,

Tomas Hrbek